# Deep Transfer Learning for Land Use and Land Cover Classification: A Comparative Study

**DOI:** 10.3390/s21238083

**Published:** 2021-12-03

**Authors:** Raoof Naushad, Tarunpreet Kaur, Ebrahim Ghaderpour

**Affiliations:** 1Accubits Invent—Artificial Intelligence R&D Lab, Accubits Technologies Inc., Trivandrum 695581, India; 2Department of Biomedical Science, Acharya Narendra Dev College, University of Delhi, Delhi 110019, India; ab235@andc.edu.du.ac.in; 3Department of Geomatics Engineering, University of Calgary, 2500 University Drive NW, Calgary, AB T2N 1N4, Canada; ebrahim.ghaderpour@ucalgary.ca

**Keywords:** land use classification, land cover classification, remote sensing, satellite imagery, EuroSAT, earth observation, deep learning, transfer learning, satellite image classification

## Abstract

Efficiently implementing remote sensing image classification with high spatial resolution imagery can provide significant value in land use and land cover (LULC) classification. The new advances in remote sensing and deep learning technologies have facilitated the extraction of spatiotemporal information for LULC classification. Moreover, diverse disciplines of science, including remote sensing, have utilised tremendous improvements in image classification involving convolutional neural networks (CNNs) with transfer learning. In this study, instead of training CNNs from scratch, the transfer learning was applied to fine-tune pre-trained networks Visual Geometry Group (VGG16) and Wide Residual Networks (WRNs), by replacing the final layers with additional layers, for LULC classification using the red–green–blue version of the EuroSAT dataset. Moreover, the performance and computational time are compared and optimised with techniques such as early stopping, gradient clipping, adaptive learning rates, and data augmentation. The proposed approaches have addressed the limited-data problem, and very good accuracies were achieved. The results show that the proposed method based on WRNs outperformed the previous best results in terms of computational efficiency and accuracy, by achieving 99.17%.

## 1. Introduction

There have been rapid advancements in remote sensing technologies. Satellite image acquisitions now take place. Unprecedented amounts of information are available, and access to data is greater. All of this allows us to understand the features of Earth more comprehensively, encouraging innovation and entrepreneurship. The enhanced ability to observe the Earth from low orbit and geostationary satellites [1] and the better spatial resolution of remote sensing data [2] have led to the development of novel approaches for remote sensing image analysis, facilitating extensive ground surface studies. Scene classification that is aimed at labelling an image according to a set of semantic categories [3] is eminent in the remote sensing field due to its extensive applications, including land use and land cover (LULC) [4,5] and land resource management [2].

Recent years have witnessed great advances in LULC classification in tasks such as denoising, cloud shadow masking, segmentation, and classification [6,7,8,9]. Extensive algorithms have been devised with concrete theoretical bases, exploiting the spectral and spatial properties of pixels. However, with an increase in the level of abstraction from pixels to objects to scenes, and the complex spatial distributions of diverse land cover types, classification continues to be a challenging task [10]. Object or pixel-based [11,12,13] approaches possessing low-level features encoding spectral, textural, and geometric properties are becoming incompetent at capturing the semantics of scenes. Hu et al. [14] deduced that more representative and high-level features, which are the abstractions of low-level features, are necessary for scene classification. Currently, convolutional neural networks (CNNs) are the dominant methods in image classification, detection, and segmentation tasks because of their ability to extract high-level feature representations to describe scenes in images [15].

Hu et al. [14] observed that in spite of CNNs’ fine ability to extract the high-level and low-level features, it is tedious to train CNNs with smaller datasets. Yin et al. [16] and Yosinski et al. [17] observed that the features learned by the layers from different datasets show common behaviour. Convolution operators from the initial layers learn the general characteristics, and towards the final layers, there is a transition to features more specific to the dataset on which the model is trained. These general and specific CNN layer feature transitions have led to the development of transfer learning [18,19]. As a result, the features learnt by the CNN model on a primary job were employed for an unrelated secondary task in transfer learning. The primary model acts as a starting point or as a feature extractor for the secondary model. The contributions made in this article are listed below.

LULC classification was performed using two transfer learning architectures, namely, the Visual Geometry Group (VGG16) and Wide Residual Networks-50 (ResNet-50), on the red–green–blue (RGB) version of the EuroSAT dataset.The performances of the methods were empirically evaluated with and without data augmentation.The model performance and computational efficiency were improved with model enhancement techniques.The RGB version of the EuroSAT dataset was benchmarked.

The rest of the paper is organised as follows. First, the related works are presented in Section 2. In Section 3, the dataset used herein is described, and the methodologies of the modified VGG16 and Wide ResNet-50 are presented. The results and analyses are demonstrated in Section 4. A discussion is presented in relation to other studies in Section 5, and finally, the paper is concluded in Section 6.

## 2. Related Works

This section mainly presents the recent studies in remote sensing scene classification using deep learning (DL) and transfer learning (TL). Furthermore, it presents the state-of-the-art image classification methods for LULC on the EuroSAT dataset.

Xu et al. [20] used principal component analysis (PCA) to reduce data redundancy, and then trained a self-organising network to classify Landsat satellite images, which outperformed the maximum likelihood method. Later, Chen et al. [21] showed the potential of DL for hyperspectral data classification with a hybrid framework which included DL, logistic regression, and PCA [22]. Stacked autoencoders were used in DL frameworks to extract high-level features. Basu et al. [5] and Zou et al. [15] used deep belief networks for remote sensing image classification and experimentally demonstrated the effectiveness of the model. Piramanayagam et al. [22] and Liu et al. [23] demonstrated the potential of CNNs for LULC classification. They actively selected training samples at each iteration with DL for a better performance. The scarcity of labelled data was tackled by implementing data augmentation techniques [24]. Furthermore, Yang et al. [25] improved the generalisation capability and performance by combining deep CNN and multi-scale feature fusion against the limited data. Liu et al. [26] also proposed a scene classification method based on a deep random-scale stretched CNN. Another constraint with remote sensing images was the presence of scenic variability, which limited the classification performance. As a work-around, the Saliency Dual Attention Residual Network (SDAResNet) was studied in [27] containing both spatial and channel attention, leading to a better performance. Later, Xu et al. [28] came up with an enhanced classification method involving the Recurrent Neural Network along with Random Forest for LULC. Another approach with an attention mechanism was studied by Alhichri et al. [29] based on the pre-trained EfficientNet-B3 CNN. They tested it on six popular LULC datasets and demonstrated its capability in remote sensing scene classification tasks. Liang et al. [30] and Pires de Lima and Marfurt [31] proposed specific fine-tuning strategies which were better than CNN for aerial image classification. Kwon et al. [32] proposed a robust classification score method for detecting adversarial examples in deep neural networks that does not invoke any additional process, such as changing the classifier or modifying input data. Bahri et al. [33] experimented with a TL technique that outperformed all the existing baseline models by using Neural Architecture Search Network Mobile (NASNet Mobile) as a feature descriptor, and also introduced a loss function that contributed to the performance.

In the context of LULC classification (Table 1) on the EuroSAT dataset, Helber et al. [34], the creators, used GoogleNet and ResNet-50 architectures with different band combinations. They found that the ResNet-50 with the RGB bands achieved the best accuracy compared to GoogleNet with the RGB bands and ResNet-50 with a short-wave infrared (SWIR) and color-infrared (CI) combination. The Deep Discriminative Representation Learning with Attention Map (DDRL-AM) method, proposed by Li et al. [35], obtained the highest accuracy of 98.74% using the RGB bands, compared to the other results listed in Table 1. Finally, Yassine et al. [36] tried out two approaches for improving accuracy of using the EuroSAT dataset. In the first approach, the 13 spectral bands of Sentinel-2 were used for feature extraction, producing 98.78% accuracy. In the second approach, 13 spectral feature bands of Sentinel-2 along with the calculated indices, such as vegetation index based on red edge (VIRE), normalised near-infrared (NNIR), and blue ratio (BR) were used for feature extraction, resulting in an accuracy of 99.58%.

## 3. Materials and Methods

TL was used to carry out the LULC classification. In past experiments, several architectures have been proposed and tested for scene classification [22,23,24]. After experimenting with and comparing different pre-trained architectures [25,26,27,28], we decided to employ VGG16 and Wide ResNet-50 for the particular use-case. The models were fine-tuned on the RGB version of the EuroSAT dataset and trained using the PyTorch framework, in the Python language. NVIDIA TESLA P100 GPUs available with Kaggle were used for model training and testing.

### 3.1. Dataset

The EuroSAT dataset is considered a novel dataset based on the multispectral image data provided by the Sentinel-2 satellite. It has 13 spectral bands consisting of 27,000 labelled and georeferenced images (2000–3000 images per class) categorised into 10 different scene classes. The image patches contain 64 × 64 pixels with a spatial resolution of 10 m. Figure 1 demonstrates some sample images from the EuroSAT dataset [34].

The RGB version of the EuroSAT dataset is used for training in this study. The labelled EuroSAT dataset is made publicly available [40]. The dataset is split into 75/25 ratios for training (20,250 images) and validation (6750 images), respectively. Mini-batches of 64 images are used for training purposes, see Data Availability Statement.

### 3.2. Transfer Learning Methods

VGG16, very deep convolutional networks, has shown that the representation depth is beneficial for the classification accuracy [41]. The pre-trained VGG model was trained on the ImageNet dataset with 1000 classes; the convolutional block possesses multiple convolutional layers. The top layers learn low-level features and the bottom layers learn high-level features of the images.

ResNet can be viewed as an ensemble of many smaller networks and has commendable performance for image recognition tasks [42,43,44]. The performance degradation problem [45] caused by adding more layers to sufficiently deep networks was tackled by ResNet via introducing an identity shortcut connection [46]. The Wide Residual Networks are an improvement over the Residual Networks. They possess more channels with increased width and decreased depth when compared to the Residual Networks [47].

In this research, the pre-trained models of VGG16 and Wide ResNet-50 were used. The VGG16 and Wide ResNet-50 pre-trained models expect input images normalised in mini-batches of 3-channel RGB images of shape (3 × H × W), where H and W are expected to be 224. The final classification layers were replaced with fully connected and dropout layers; see Figure 2. ReLU and log-softmax activation functions were also used. The initial layers from training were frozen, and the modified layer was fine-tuned with the EuroSAT dataset. The model was trained for 25 epochs with a batch size of 64. Adam [48] was used as the model optimiser with categorical cross-entropy loss for loss calculation. To enhance the model’s efficiency in terms of computation time and performance, model enhancement techniques such as gradient clipping, early stopping, data augmentation, and adaptive learning rates were used.

### 3.3. Model Performance Enhancement Methods

#### 3.3.1. Data Augmentation

The diversity and volume of training data play eminent roles in training a robust DL model. Basic data augmentation techniques [49] enhance the diversity of the data to some extent by introducing visual variability, which helps the model to interpret the information with more accuracy. For the EuroSAT dataset, the data augmentation techniques used were Gaussian blurring, horizontal flip, vertical flip, rotation, and resizing. There are many data augmentation techniques available, but due to the inherent uniformity in the EuroSAT dataset, most of the data augmentation techniques did not have a significant impact.

#### 3.3.2. Gradient Clipping

Gradient clipping [50] can prevent vanishing and exploding gradient issues that mess up the parameters during training. In order to match the norm, a predefined gradient threshold is defined. Gradient norms that surpass the threshold are reduced to match the norm. The norm is calculated over all the gradients collectively, and the maximum norm is 0.1.

#### 3.3.3. Early Stopping

Early stopping is a regularisation technique for deep neural networks which stops the training after an arbitrary number of epochs once the model performance stops improving on a held-out validation dataset. In essence, throughout training, the best model weights are saved and updated. When parameter changes no longer provide an improvement (after a certain number of iterations), training is terminated and the last best parameters are utilised (Figure 3). This process reduced overfitting and enhanced the generalisation capability of deep neural networks.

#### 3.3.4. Learning Rate Optimisation

The learning rate is a hyperparameter that controls how much the model weights are updated in response to the anticipated error in each iteration. Choosing the learning rate may be difficult, since a value that is too small can lead to a lengthy training procedure with significant training error, whereas a value too big can lead to learning a sub-optimal set of weights too quickly (without reaching the local minima) or an unstable training process [51]. To reduce the learning rate, ReduceLROnPlateau was used [52]. When learning becomes static, models frequently benefit from reducing the learning rate by a factor of 2–10. The learning rate was lowered by a factor of 0.1 with patience (number of epochs with no improvement) as 2. Adam was used as the optimiser with the maximum learning rate of 0.0001.

## 4. Results

In this section, the results are separately demonstrated for the two different transfer learning approaches employed for the study. For training each model, all the hyperparameters were finalised by preliminary experiments. The models were trained with a 75/25 split for training and testing, respectively. In other words, the models were trained on 75% of data (randomly selected) and tested on the other 25%. Similarly, five different such sets were used for evaluation. Data augmentation is implemented to increase the effective training set size.

### 4.1. VGG16—Visual Geometry Group Network

The EuroSAT dataset on VGG16 architecture was fine-tuned by freezing the top layers and training only the added classification layers (Figure 2a) with different hyperparameters. The pre-trained weights had the advantage of the learnings that they achieved on the ImageNet dataset.

While training without data augmentation (WDA), a validation accuracy of 98.14% was achieved, whereas training with data augmentation resulted in a better accuracy of 98.55% (Table 2). The early stopping method was used with patience of 5, and the model with the highest validation accuracy was saved. This approach helped in preventing the overfitting of the model and saved computational time. Due to early stopping, the training stopped at the 21st epoch (18th—WDA); the total number of epochs was 25. It took 2 h, 4 min, 12 s to train 21 epochs, which means approximately 6.1 min for each epoch. However, without data augmentation, it took 1 h, 47 min, 24 s to train 18 epochs, which means approximately 5.9 min for each epoch (Table 2).

Figure 4 shows the training and validation loss and accuracy diagrams. It can be seen that in the first epoch, both the loss and accuracy improved exponentially and then showed a linear relation in epochs 2–10. During this period, some instability in learning was observed, and towards the end, no significant improvement was noticed. Since an adaptive learning rate with ReduceLROnPlateau was used, the learning rate was updated thrice during the training, which certainly helped the model to achieve the optimum result.

### 4.2. Wide ResNet-50—Wide Residual Network

In the first approach of training WDA, the model was able to achieve a validation accuracy of 99.04%, which was outperformed by the approach with data augmentation, which achieved an accuracy of 99.17% (Table 2). Hence, the model with the best performance was considered. With early stopping, the training stopped at the 23rd epoch (total 25 epochs), whereas WDA training stopped at the 14th epoch. The best model took 2 h, 7 min, 53 s to run 23 epochs with 5.6 min per epoch, which was better than the VGG16 (Table 2).

The loss and accuracy graphs showed steady learning in the first epoch (Figure 5). Towards the 15th epoch, the learning showed almost a linear relationship with some instability in between. Furthermore, between the 15th and 23rd epochs, delayed and little learning was achieved because of the updating of the learning rate to smaller optimum values to calculate the best result. The learning rate changed thrice in the entire training period.

Figure 6a demonstrates the confusion matrix of VGG16, based on validation data, which shows the class-wise performance of the model. The Forest, Highway, Residential, and Sea/Lake classes showed the best performance—above 99% accuracy. Permanent Crop, Herbaceous Vegetation, and Pasture seem to have been predicted most poorly. Annual Crop, Permanent Crop, Pasture, and Herbaceous Vegetation were misclassified because of the similarity in topological features. By analysing the images of these classes, it was understood that they share common features that might confuse the model.

Figure 6b shows the confusion matrix for Wide ResNet-50. The Forest and Sea/Lake classes were predicted most accurately: 99.86%. The class Permanent Crop was predicted least accurately: 97.39%. There was improved accuracy and fewer misclassifications for all classes except River. Figure 7 demonstrates some of the correct predictions using Wide Resnet-50 and also shows a River scene that was incorrectly predicted as Highway (see the middle panel). The modified VGG16 also incorrectly predicted this River scene as Highway and predicted the Permanent Crop scene, shown in the top-middle panel in Figure 7, as Herbaceous Vegetation.

## 5. Discussion

In this study, the challenge of LULC classification was addressed using deep transfer learning techniques. For this task, two prominent transfer learning architectures, namely, VGG16 and Wide ResNet-50, were used on the EuroSAT dataset. Focusing on the LULC classification of the RGB bands of the EuroSAT dataset, a state-of-the-art accuracy of 99.17% was achieved by using Wide ResNet-50.

Experimentally, the best fine-tuning parameters were found for VGG16 and Wide ResNet-50 with the RGB bands of the EuroSAT dataset. The parameters that contributed to the best performance were used to create the final models. The models were compared with and without data augmentation. Both of these architectures were compared based on computational training time, the number of epochs trained, and test data accuracy (Table 2). From the results, it was observed that Wide ResNet-50 was computationally more feasible, as the time taken for each epoch to train was less than that of VGG16, even though the former is a deeper network.

The number of epochs trained was less without data augmentation due to early stopping and limited data. The model converged early and did not show much improvement, hence consuming a shorter training time. In contrast, more epochs were used with data augmentation because it generated more data for the model to learn the features from, which provided better generalisation and ultimately led to a better accuracy. With more high-resolution data, the architecture proposed herein can create and learn more adversarial examples [53] and make better predictions.

From the confusion matrix shown in Figure 6b, one can see that the Forest class, followed by the Sea/Lake class, was the best handled, as it was hardly misclassified. Similarly, due to similar topological features, Herbaceous Vegetation, Annual Crop, Pasture, and Permanent Crop classes were confused. The Highway class was misclassified as the River class because of a similar linear appearance. A similar trend was observed in the VGG16 confusion matrix (Figure 6a). The presence of clear and distinct topological features for the Forest and Sea/Lake classes, i.e., mostly green and blue for both the images, led to accurate results. Similarly, Pasture, Herbaceous Vegetation, and Annual Crops were misclassified to higher degrees. Again, the Highway and River classifications were also confused because of similar topological features. Thus, from these trends, it was concluded that the model’s training was mimicking human learning patterns. With the presence of more inter and intra-class variability in the dataset, these faulty learning patterns could be significantly improved. Another effective approach would be incorporating the invisible bands, such as near-infrared, into the models for distinguishing between road and river [7,36]. From the feature understanding capability depicted by the confusion matrices of both the models, the learning pattern of the architectures was found to be quite comparable. The major difference lay only on how well the model was understanding everything, i.e., the classification accuracy.

In this research, the performances of Wide ResNet-50 and VGG16 with multiple validation datasets were intensively compared. The prediction accuracy of Wide ResNet-50 on the EuroSAT dataset was found to be better than of VGG16 by at least 0.6% of the total validation set. As mentioned in Table 2, the best performing model of Wide ResNet-50 achieved 99.17%, whereas it was 98.55% for VGG16. Thus, it was understood that Wide ResNet-50 performed better than VGG16. As shown in Table 1, the accuracy of 99.17% achieved using Wide ResNet-50 with the RGB bands was higher than the highest accuracy of 98.74% achieved using the DDRL-AM model with RGB bands.

## 6. Conclusions

The objective of this article was to investigate how the transfer learning architectures for LULC classification perform. The study was based on two potential architectures, namely, VGG16 and Wide ResNet-50, fine-tuned with RGB bands of the EuroSAT dataset for the classification. Much like the findings in other experiments, it was found that transfer learning is a quite reliable approach that can produce the best overall results. The proposed methodology improved the state-of-the-art and provided a benchmark with an accuracy of 99.17% for the RGB bands of the EuroSAT dataset.

The classification results prior to and after data augmentation were compared. Data augmentation techniques elevated the diversification of the dataset, as they only increased the visual variability of each training image without generating any new spectral or topological information. Evidently, the experimental results with data augmentation outperformed those from the same model architecture trained on the original dataset. Model enhancement techniques such as regularisation, early stopping, gradient clipping, learning rate optimisation, and others were implemented to make the model training more efficient, improve the performance, and ultimately reduce the computational time required. The Wide ResNet-50 architecture was found to generate better results than VGG16, though the same data augmentation approaches were applied to both. Even though Wide ResNet-50 produced better results, the learning patterns of the models were similar; the only difference was found in the accuracy of the class predictability.

This problem may be solved by supplementing the quality and quantity of data. The generation of datasets with higher inter and intra-class variability, supported by robust deep learning architectures with data augmentation techniques, could effectively increase the representational power of the deep learning network. Thus, the proposed methodology is an effective exploitation of the satellite datasets available and deep learning approaches to achieve the best performance. The applications can be extended to multiple real-world Earth observation applications for remote sensing scene analysis.

## Figures and Tables

**Figure 1 sensors-21-08083-f001:**
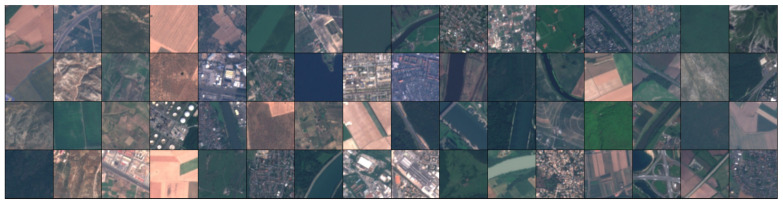
EuroSAT dataset sample images. The available classes are Forest, Annual Crop, Highway, Herbaceous Vegetation, Pasture, Residential, River, Industrial, Permanent Crop, and Sea/Lake.

**Figure 2 sensors-21-08083-f002:**
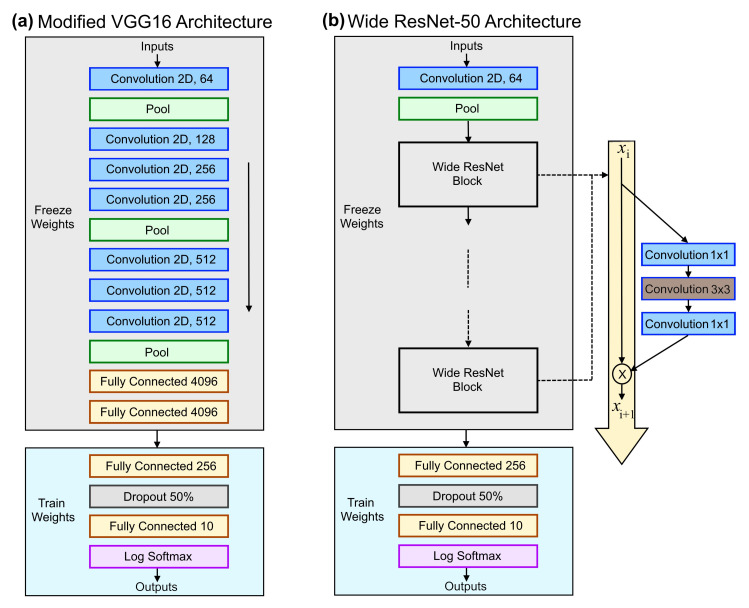
Model architectures: (**a**) modified VGG16 architecture with training and freezing layers, and (**b**) wide ResNet-50 architecture with training and freezing layers.

**Figure 3 sensors-21-08083-f003:**
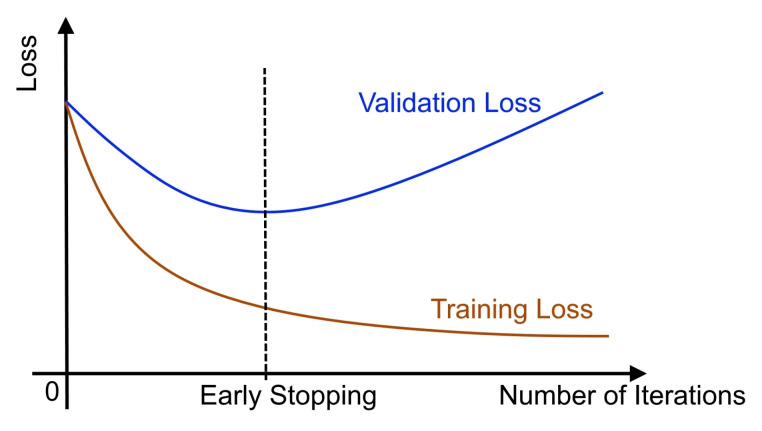
Early stopping: training is stopped as soon as the performance on the validation loss stops decreasing even though the training loss decreases.

**Figure 4 sensors-21-08083-f004:**
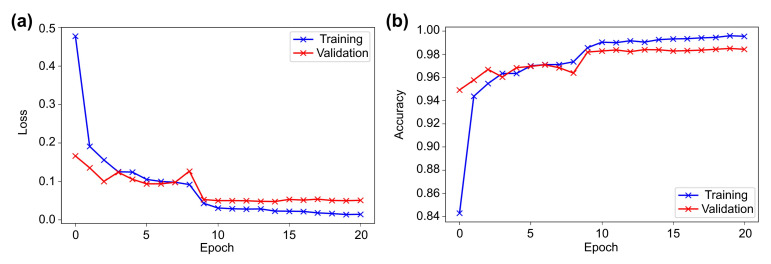
The VGG16 results representing the history of training and validation (**a**) loss and (**b**) accuracy across the epochs.

**Figure 5 sensors-21-08083-f005:**
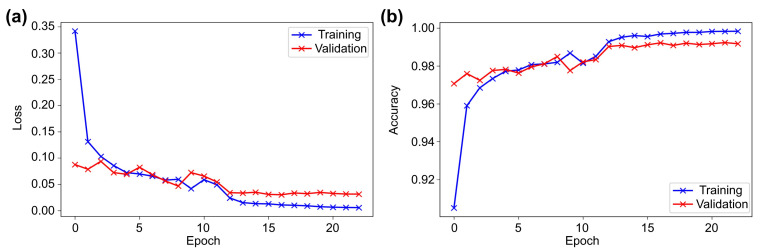
The Wide ResNet-50 results representing the history of training and validation (**a**) loss and (**b**) accuracy across the epochs.

**Figure 6 sensors-21-08083-f006:**
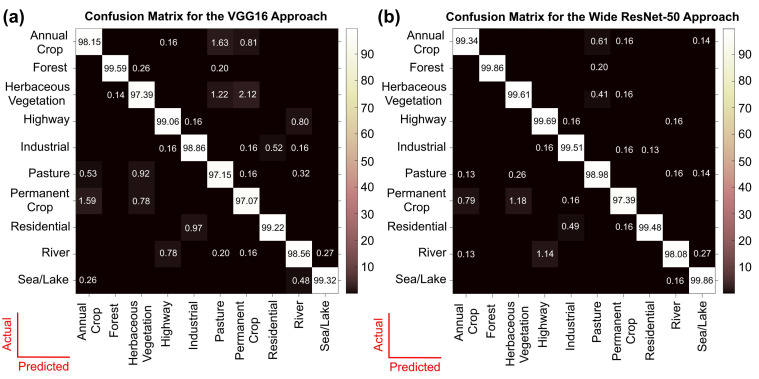
The confusion matrices for the (**a**) VGG16 and (**b**) Wide ResNet-50 architectures applied to the EuroSAT dataset.

**Figure 7 sensors-21-08083-f007:**
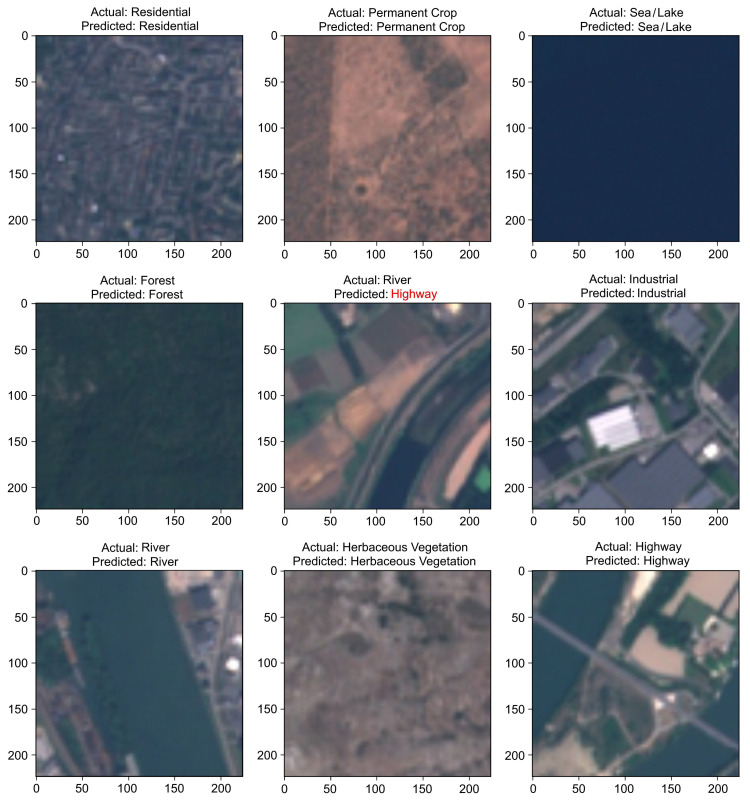
The Wide ResNet-50 sample results. Shown are the actual and predicted values of sample inputs from the test dataset. Note that VGG16 also predicted these scenes the same as Wide ResNet-50, but incorrectly predicted the top-middle scene as Herbaceous Vegetation.

**Table 1 sensors-21-08083-t001:** Comparative analysis of studies for LULC classification with the EuroSAT dataset.

Authors	Model	Bands	Accuracy
Helber et al. [34]	GoogleNet	RGB	98.18%
Helber et al. [34]	ResNet-50	SWIR	97.05%
Helber et al. [34]	ResNet-50	CI	98.30%
Helber et al. [34]	ResNet-50	RGB	98.57%
Chen et al. [37]	Knowledge distillation	RGB	94.74%
Chong [38]	VGG16	RGB	94.50%
Chong [38]	4-convolution max-pooling layer	All 13 spectral bands	94.90%
Sonune [39]	Random Forest	RGB	61.46%
Sonune [39]	ResNet-50	RGB	94.25%
Sonune [39]	VGG19	RGB	97.66%
Li et al. [35]	DDRL-AM	RGB	98.74%
Yassine et al. [36]	CNN	All 13 spectral bands	98.78%
Yassine et al. [36]	CNN	All 13 spectral bands + VIRE + NNIR + BR	99.58%

**Table 2 sensors-21-08083-t002:** Comparative experimental results of VGG16 and Wide ResNet-50 with and without data augmentation.

Model	Epochs Trained	Total Time	Time Per Epoch	Accuracy
VGG16 (Without Data Augmentation)	18	1 h 47 min 24 s	5.9 min	98.14%
VGG16 (With Data Augmentation)	21	2 h 4 min 12 s	6.1 min	98.55%
Wide ResNet-50 (Without Data Augmentation)	14	1 h 19 min 48 s	5.5 min	99.04%
Wide ResNet-50 (With Data Augmentation)	23	2 h 7 min 53 s	5.6 min	99.17%

## Data Availability

Data associated with this research are available online. The EuroSAT dataset is freely available for download [40]. The Jupyter Notebooks used for the training the image classifier is available for download at https://github.com/raoofnaushad/EuroSAT_LULC (accessed on 24 October 2021).

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
