# Peer review of "Deep Transfer Learning for Land Use and Land Cover Classification: A Comparative Study"

_sensors, 2021, doi:10.3390/s21238083_

Round 1

Reviewer 1 Report

The manuscript presented a comparative study on two transfer learning architectures: VGG16 and Wide ResNet-50 with the EuroSAT dataset. The manuscript was presented clearly and logically. The method and results had no obvious flaw, but the conclusion was not persuasive enough. It was hard to conclude the ResNet-50 performed better than VGG16, because the accuracies of ResNet-50 and VGG16 were almost at the same level with or without augmentation, which were only shown in very slight difference. So maybe intensive comparison and analysis should be conducted to provide more strong evidence to make the conclusion.

Author Response

Dear Reviewer 1,

We would like to thank you very much for your time and for providing very valuable comments that helped us improve the presentation of our manuscript. Below, please find our response to your comment in red that is also highlighted in the manuscript.

The manuscript presented a comparative study on two transfer learning architectures: VGG16 and Wide ResNet-50 with the EuroSAT dataset. The manuscript was presented clearly and logically. The method and results had no obvious flaw, but the conclusion was not persuasive enough. It was hard to conclude the ResNet-50 performed better than VGG16, because the accuracies of ResNet-50 and VGG16 were almost at the same level with or without augmentation, which were only shown in very slight difference. So maybe intensive comparison and analysis should be conducted to provide more strong evidence to make the conclusion.

Response.

Thank you very much for pointing this out. To clarify this better, we have modified the last paragraph of the Discussion part as follows:

“In this research, the performance of Wide ResNet-50 and VGG16 with multiple validation dataset was intensively compared. The prediction of Wide ResNet-50 on the EuroSAT dataset was found better than VGG16 by at least 0.6% of the total validation dataset. As mentioned in Table 2, the best performing model of Wide ResNet-50 was 99.17%, while it was 98.55% for VGG16. Thus, it was understood that Wide ResNet-50 performed better than VGG16. From Table 1, the achieved accuracy of 99.17% using Wide ResNet-50 with the RGB bands is higher than the highest achieved an accuracy of 98.74% using the DDRL-AM model with RGB bands.”

Please note that for multiple validation set, we trained the models on random 75% data and tested on the other 25%. Similarly, we had 5 different such sets for evaluation.

We hope that the changes we made are satisfactory. Please kindly let us know if you have any further comments.

Thank you

Best regards,

The authors

Reviewer 2 Report

• This paper deals with an exciting topic. The article has been read carefully, and some crucial issues have been highlighted in order to be considered by the author(s).

• All the acronyms should be defined and explained first before using them such that they become evident for the readers.

• Most of the typos and incorrect grammars have been corrected, but it is still necessary to subject the paper to proofreading.

• The paper needs to be restructured in order to be precise.The Introduction and related work parts give valuable information for the readers as well as researchers. In addition recent papers should be added in the part of related work.

• As it is real time application oriented, authors should care over the outcome of the proposed framework by meeting the future requirements too.

• Representation of figures needs to be improved.

• Grammatical errors should be validated.

• It would be good if similar domains, such as adversarial examples, would be reflected in future research or related work.

[1] Kwon, Hyun, et al. "Classification score approach for detecting adversarial example in deep neural network." Multimedia Tools and Applications 80.7 (2021): 10339-10360.

Author Response

Dear Reviewer 2,

We would like to thank you very much for your time and for providing very valuable comments that helped us improve the presentation of our manuscript. Below, please find our responses to your comments in red that are also highlighted in the manuscript.

This paper deals with an exciting topic. The article has been read carefully, and some crucial issues have been highlighted in order to be considered by the author(s).

[1] All the acronyms should be defined and explained first before using them such that they become evident for the readers.

Response. We have carefully checked the acronyms and defined them, the first time they appear in the text and abstract. We also updated our acronym table at the end of the manuscript.

[2] Most of the typos and incorrect grammars have been corrected, but it is still necessary to subject the paper to proofreading.

Response. We have carefully proofread the article and found some grammar/typo issues and corrected. We also use only passive voice as per reviewer 3 suggestions.

[3] The paper needs to be restructured in order to be precise. The Introduction and related work parts give valuable information for the readers as well as researchers. In addition, recent papers should be added in the part of related work.

Response. We have structured the paper according to the MDPI guidelines: 1. Introduction, 2. Related Works, 3. Materials and Methods, 4. Results, 5. Discussion, 6. Conclusions. All recent papers related to the Eurosat dataset are included. We also appreciate your suggested reference, and we included it in the Related Works.

[4] As it is real time application oriented, authors should care over the outcome of the proposed framework by meeting the future requirements too.

Response. We appreciate your comment. However, this paper is written from a comparative study perspective but not as a real time application study. The real time application is subject to the future work.

[5] Representation of figures needs to be improved.

Response. All the figures are professionally prepared with a resolution of at least 300dpi, and in the Result section they are carefully structured.

[6] Grammatical errors should be validated.

Response. We have carefully proofread the article and corrected the grammar/typo issues that we found. We also used only passive voice throughout the entire manuscript.

[7] It would be good if similar domains, such as adversarial examples, would be reflected in future research or related work.

[1] Kwon, Hyun, et al. "Classification score approach for detecting adversarial example in deep neural network." Multimedia Tools and Applications 80.7 (2021): 10339-10360.

Response. Thank you for suggesting this nice work. We have included it in the Related Works section.

“Kwon et al. [32] proposed a robust classification score method for detecting adversarial examples in deep neural networks that does not invoke any additional process, such as changing the classifier or modifying input data.”

Also, we are already addressing the adversarial examples problem with multiple data augmentation techniques. With more high-resolution data our architecture can create and learn more adversarial examples and make better predictions.

Thus, In Discussion, we also added:

“Furthermore, with more high-resolution data, the proposed architecture can create and learn more adversarial examples [53] and make better predictions.”

The new references added are:

[32] Kwon, H.; Kim, Y.; Yoon, H.; Choi, D. Classification score approach for detecting adversarial example in deep neural network. Multimed Tools Appl 2021, 80, 10339--10360. https://doi.org/10.1007/s11042-020-09167-z

[53] Kwon, H.; Lee, J. Diversity Adversarial Training against Adversarial Attack on Deep Neural Networks. Symmetry 2021, 13, 428. https://doi.org/10.3390/sym13030428

All the changes mentioned above are also highlighted in the revised draft.

We hope that the changes we made are satisfactory. Please kindly let us know if you have any further comments.

Thank you

Best regards,

The authors

Reviewer 3 Report

This paper is excellently structured, readable and innovative. I definitely suggest acceptance for publication, but please do the following minor changes.

Parts of this paper were written in passive and part of it in active form. Please, be consistent. Please add a slash between Land Use Land Cover, like Land Use/Land Cover wherever it appears in the paper. I find it unnecessary to put the same information both in the table (Table 1.) and text section (lines 98-111).

Specific comments:

lines 113, 126, 142: all sections starts with the word “Herein“

Author Response

Dear Reviewer 3,

We would like to thank you very much for your time and for providing very valuable comments that helped us improve the presentation of our manuscript. Below, please find our responses to your comments in red that are also highlighted in the manuscript.

This paper is excellently structured, readable and innovative. I definitely suggest acceptance for publication, but please do the following minor changes.

Parts of this paper were written in passive and part of it in active form. Please, be consistent.

Response. Agreed. We used the passive voice and removed the words “We and Our”.

Please add a slash between Land Use Land Cover, like Land Use/Land Cover wherever it appears in the paper.

Response. Agreed. However, since we are doing Land Use (Highways, industrial, etc.) and Land Cover (River, Forest, Vegetation, etc.) classification simultaneously, we now write this phrase as “Land Use and Land Cover” instead of Land Use/Land Cover because slash can also mean “or” instead of “and”. Also, Land Use and Land Cover is the common phrase used by other authors in our reference list.

I find it unnecessary to put the same information both in the table (Table 1) and text section (lines 98-111).

Response. Agreed. we removed the redundant texts that were repeated in Table 1 but kept some of them because it provides further information and defines the acronyms.

Specific comments:

lines 113, 126, 142: all sections starts with the word “Herein“

Line 126. We removed Herein and rephrase the sentence as “The RGB version of the EuroSAT dataset is used for training in this study.”

Line 142. We replaced “Herein” with “In this research”.

These changes are highlighted in the revised draft.

We hope that the changes we made are satisfactory. Please kindly let us know if you have any further comments.

Thank you

Best regards,

The authors

Round 2

Reviewer 1 Report

The revised version has answered my question and I suggest the paper to be published in the journal.

Reviewer 2 Report

This paper is worth for acceptance